# *Cutibacterium acnes* KCTC 3314 Growth Reduction with the Combined Use of Bacteriophage PAP 1-1 and Nisin

**DOI:** 10.3390/antibiotics12061035

**Published:** 2023-06-10

**Authors:** Min-Hui Han, Shehzad Abid Khan, Gi-Seong Moon

**Affiliations:** 1Major of Biotechnology, Korea National University of Transportation, Chungju 27909, Republic of Korea; qhtjrrhdwn10@naver.com; 2Atta-ur-Rahman School of Applied Biosciences (ASAB), National University of Sciences and Technology (NUST), Islamabad 44000, Pakistan; shehzadkhan12786@gmail.com; 34D Convergence Technology Institute, Korea National University of Transportation, Chungju 27909, Republic of Korea

**Keywords:** bacteriophage, *Cutibacterium acnes*, *Lactococcus lactis*, bacteriocin, skin acne

## Abstract

Severe acne has high psychological impacts recorded worldwide, from depression to suicide. To control acne infection, bacteriophage could be used in synergy or combination with antibiotics/antimicrobials. Bacteriophages are specific to their hosts without interfering with the normal skin microbes and have the ability to lyse bacterial cells. In this current study, the bacteriophage PAP 1-1 was isolated, characterized, and tested against the pathogenic acne-causing bacterium *Cutibacterium acnes*. Examination under transmission electron microscopy (TEM) revealed that the newly isolated phage has a morphology typical of siphoviruses. Phylogenetic analysis, utilizing the maximum likelihood (ML) algorithm based on complete genome sequences, revealed that PAP 1-1 clustered together with bacteriophages active to *Propionibacterium acnes* (now known as *C. acnes*), forming a distinct evolutionary lineage. The genomic analysis further identified the presence of an endolysin gene in PAP 1-1, suggesting its potential to regulate the growth of *C. acnes*. Subsequent experiments conducted in RCM broth confirmed the ability of PAP 1-1 to effectively control the proliferation of *C. acnes*. In combination with bacteriocin from *Lactococcus lactis* CJNU 3001 and nisin, PAP 1-1 greatly decreased the viable cell counts of *C. acnes* in the broth.

## 1. Introduction

Healthy human skin harbor diverse skin microbiota, mainly belonging to the genera *Corynebacterium*, *Staphylococcus,* and *Cutibacterium*, many of which act as commensals or opportunistic pathogens [1,2]. A balance between members of these genera is essential to maintain healthy skin conditions [3,4]. Among the skin microbiota, some *Cutibacterium acnes* strains are commensal, helping to maintain skin homeostasis by avoiding the colonization of dangerous pathogens, while other strains are opportunistic pathogens, primarily responsible for producing invasive infections that result in inflammatory acne on the skin [4,5,6,7]. The global prevalence of acne infection among the human population is estimated to be approximately 9%, with infections occurring across all age groups without any specific demographic preference [8]. Notably, 85% of individuals affected by acne fall within the age range of 12 to 24 years old. Severe acne has been consistently associated with significant psychological impacts on a global scale, ranging from symptoms of depression to an increased risk of suicide. Extensive documentation exists regarding the detrimental effects on mental well-being caused by severe acne, highlighting the need for developing therapies and interventions for individuals affected by this condition [9]. Several virulence factors in different strains of *C. acnes*, including neuraminidase, polyunsaturated fatty acid isomerase, lipase, and heat shock and iron acquisition proteins, are involved in inflammatory potential [4]. Besides these virulence factors, *C. acnes* strains are involved in biofilm formation and possess resistance to clindamycin, erythromycin, and tetracycline [10,11,12]. These concerns limit the use of antibiotics for acne management and drive researchers towards finding alternative treatments for *C. acnes* strains causing acne infection.

Phage therapy, which involves the use of bacteriophages (viruses that infect bacteria) for the treatment of bacterial infections, has garnered significant attention among researchers as a potential adjunct to antibiotics [13]. In phage therapy, phages are used alone or in combination with other phages and/or drugs to target bacterial strains [13]. Phage therapy has demonstrated effectiveness in controlling infections caused by specific bacteria, with the advantage of phages being safe for humans and environmentally friendly [14,15]. Bacteriophages are ubiquitously present in the environment, ranging from unicellular to multicellular organisms, and play a role in regulating ecosystems and controlling bacterial populations [16,17,18]. Previous research has explored how phages help in maintaining the human microbiota against pathogenic bacteria [16,19,20,21,22]. Recently, the therapeutic potential of bacteriophages has been evaluated against specific pathogenic bacteria to control their proliferation. For instance, bacteriophages VP01 and pVa-21 have successfully reduced the growth of *Vibrio alginolyticus* [13]. Additionally, bacteriophages can play a crucial role in controlling outbreaks related to biofilms by infecting bacteria in the outermost regions of the matrix and spreading within the complex biofilm, where antibiotics often fail to penetrate [13]. Therefore, phages have been considered as alternatives to antibiotics, particularly in eradicating biofilms.

Apart from bacteriophages, bacteria themselves produce various other antimicrobial substances, including bacteriocins, organic acids, hydrogen peroxide, and low-molecular-weight compounds such as fatty acids and reuterin [23]. Bacteriocins are antimicrobial peptides synthesized by bacterial ribosomes and possess inhibitory activity, often exhibiting a narrow spectrum but occasionally displaying broad-spectrum effects [24]. Bacteriocins have been extensively studied as bio-preservatives in food against pathogenic bacteria, with commercially available examples such as nisin, pediocin, lacticin 3147, enterocins, plantaricins, and sakacin [25]. Each bacteriocin may have different target spectra and modes of action against specific bacteria. Pediocin is produced by *Pediococcus* species, such as *Pediococcus acidilactici* and *Pediococcus pentosaceus* [26]. It exhibits antimicrobial activity against a range of Gram-positive bacteria, including *Listeria* spp. and *Clostridium perfringens*. Lacticin 3147 bacteriocin produced by *Lactococcus lactis* subsp. *lactis* and has potent antimicrobial activity against several Gram-positive bacteria, including *Staphylococcus aureus* and *Streptococcus pneumoniae* [27]. Enterocins are a group of bacteriocins produced by *Enterococcus* species [28]. They display activity against various Gram-positive bacteria, including *Enterococcus faecalis*, *Listeria monocytogenes*, and *Clostridium difficile*. Plantaricins are bacteriocins produced by *Lactobacillus plantarum* [29]. They exhibit antimicrobial activity against a wide range of Gram-positive bacteria, including *Bacillus cereus*, *Staphylococcus aureus*, and *Listeria monocytogenes*. Sakacin is a bacteriocin produced by *Lactobacillus sakei* and has been shown to have antimicrobial activity against *Listeria monocytogenes*, making it valuable for food preservation [30].

Nisin is a well-known bacteriocin produced by *Lactococcus lactis* [31]. The bacterium produces nisin as a defense mechanism against other competing bacteria in its environment. Nisin acts by disrupting the integrity of target bacterial cell membranes, leading to cell death. It primarily targets Gram-positive bacteria, including foodborne pathogens such as *Listeria monocytogenes* and *Staphylococcus aureus*, as well as acne-causing bacteria such as *C. acnes*, by binding to specific lipid II molecules, a critical component in the cell wall synthesis process. This ultimately causes the formation of pores and leakage of intracellular contents. Nisin possesses several desirable properties that make it an effective antimicrobial agent. It is heat-stable, pH-resistant, and exhibits a broad spectrum of activity against a wide range of bacteria. One study in 2013 evaluated the efficacy of nisin against various strains of *C. acnes*. The researchers found that nisin exhibited antimicrobial activity. They concluded that nisin could be a potential alternative treatment for acne, particularly in cases where antibiotic resistance is a concern [32]. Another study in 2017 investigated the antibacterial effects of nisin against *C. acnes* biofilms. The researchers observed that nisin effectively reduced the viability of the biofilms, suggesting its potential as a topical treatment for acne.

In this study, we demonstrate the efficacy of phage therapy in reducing the growth of *C. acnes*. Our results provide compelling evidence that the phage alone, as well as in combination with nisin, exhibits high effectiveness against *C. acnes* KCTC 3314, which was isolated from acne lesions on human facial skin. Additionally, we present evidence of the broad spectrum activity of a crude bacteriocin derived from the *L. lactis* CJNU 3001 strain, as well as commercially available nisin. Both the crude bacteriocin and nisin were compared and tested against the *C. acnes* KCTC 3314 strain, further supporting their potential as effective treatments against *C. acnes* infections.

## 2. Results

### 2.1. Isolation and Morphology of PAP 1-1

The morphology of phage plaques was observed on the lawn of *C. acnes* KCTC 3314 after 48 h of anaerobic incubation of the bacteriophage PAP 1-1 (Figure 1A). After the purification of the bacteriophage PAP 1-1, phage particle morphology was observed under a transmission electron microscope (Figure 1B). The bacteriophage PAP 1-1 had a morphology similar to that of siphovirus [7] and showed better growth on RCM than on the BHI medium (Figure 1C).

### 2.2. Genome Features

Using Prokka PAP 1-1, genome annotations were performed (Table 1). The de novo assembly of the genome-sequencing data of PAP 1-1 produced a single circular contig of 29,534 bp in size and consisting of a total of 40 protein-coding genes. Among them, 26 and 4 genes involved in head and tail morphogenesis and 4 genes involved in lysis were identified. Based on this single contig, the G + C content of PAP 1-1 was calculated as 54.0%. The genome sequence of PAP 1-1 was deposited in the GenBank database (accession number OP491959).

### 2.3. Phylogenetic Analysis and Genomic Comparison

The phylogenetic analysis using the ML algorithm based on whole-genome sequences showed that PAP 1-1 formed a close phyletic lineage with *P. acnes* ATCC29399BC (Figure 2) and formed a distinct phyletic lineage with other phages from *Propionibacterium* genus. Phylogenetic analysis results showed that PAP 1-1 belongs to bacteriophages isolated from *P. acnes*.

### 2.4. Reduction in the Growth of C. acnes KCTC 3314 with Bacteriophage PAP 1-1

The antibacterial efficacy of PAP 1-1 was assessed against *C. acnes* KCTC 3314 in a dose-dependent manner over a period of 12 and 24 h of incubation. The 0.001 MOI (multiplicity of infection as plaque-forming units (PFU)/CFU for host) effectively suppressed the growth of *C. acnes* KCTC 3314, resulting in a 1.6 log drop in CFU/mL compared with the control group (Figure 3). Notably, 0.01 MOI led to a substantial reduction in the cell population of *C. acnes* KCTC 3314, resulting in a two-log drop in CFU/mL. Gradually increasing the bacteriophage PAP 1-1 from 0.1 to 1 MOI effectively controlled the growth of *C. acnes* KCTC 3314 during the initial 12 h of incubation, which then gradually increased to 6.5 CFU/mL, similar to the count observed with 0.01 MOI after 24 h of incubation. The results show that a specific concentration of PAP 1-1 controls the growth of *C. acnes* KCTC 3314.

### 2.5. Reduction in the Growth of C. acnes KCTC 3314 Followed by Co-treatment with Bacteriophage PAP 1-1 and Bacteriocin

The antibacterial activity of the crude bacteriocin from *L. lactis* CJNU 3001 was tested against *C. acnes* KCTC 3314 alone and also in combination with 0.01 MOI in a dose-dependent manner after 12 and 24 h of incubation (Figure 4A,B). The bacteriocin treatment against *C. acnes* KCTC 3314 greatly reduced its growth, and cell counts gradually decreased with an increase in bacteriocin from 6.25 to 50 AU/mL (Figure 4A). The cell counts of *C. acnes* KCTC 3314 were reduced after treatment with bacteriocin ranging between 12.5 and 50 AU/mL, specifically with 25 AU/mL and 50 AU/mL of bacteriocin in which 5 log CFU/mL difference in growth levels were observed as compared to control. In the co-treatment of 0.01 MOI and the bacteriocin, the cell counts of *C. acnes* KCTC 3314 were reduced to a certain level after 24 h of incubation, and a significant reduction was observed with the co-treatment of 0.01 MOI and 50 AU/mL of bacteriocin (Figure 4B).

### 2.6. Reduction in the Growth of C. acnes KCTC 3314 Followed by Co-treatment with Bacteriophage PAP 1-1 and Nisin

In our previous study, the partial whole-genome sequencing of *L. lactis* CJNU 3001 (GenBank accession no. JAIZVR000000000.1) was found to harbor a corresponding structural gene for nisin. The activity of nisin against *C. acnes* KCTC 3314 was tested alone and in co-treatment with bacteriophage PAP1-1 using the same procedure as described above for the bacteriocin (Figure 5A,B). Nisin concentration range between 12.5 IU/mL and 25 IU/mL reduced the cell counts to a certain level, compared with 6.25 IU/mL and 3.125 IU/mL of nisin and control (Figure 5A). In co-treatment with 0.01 MOI, the results remained almost the same as those observed in the case of nisin alone except for the combination of 0.01 MOI and 6.25 IU/mL of nisin in which a notable reduction in the number of cells of *C. acnes* KCTC 3314 was observed compared with nisin alone treatment (Figure 5B). The concentration of 3.125 IU/mL was found to be ineffective against *C. acnes* KCTC 3314 alone and also in combination with PAP 1-1 (Figure 5A,B).

## 3. Discussion

The skin’s microbial community plays a crucial role in preventing pathogen invasion and the formation of biofilms. However, the exact impact of bacteriophages on modulating bacterial communities is not well understood. In this study, we focused on isolating, characterizing, and studying the bacteriophage PAP 1-1. Our research significantly enhanced our understanding of PAP 1-1’s activity against *C. acnes* KCTC 3314, a bacterium isolated from acne lesions on human facial skin. We examined the effects of PAP 1-1 alone and in combination with other antimicrobial compounds, including a crude bacteriocin from *L. lactis* CJNU 3001 and nisin. Morphological characterization using TEM revealed that PAP 1-1 exhibited siphovirus-like morphology, possessing an icosahedral head and a long tail [33]. Clear plaques against *C. acnes* KCTC 3314 on the RCM agar plate and the absence of lysogenic repressor genes, as determined via BLAST search, indicated that PAP 1-1 is a lytic bacteriophage [14,34]. Previous studies reported bacteriophage activity against 88% of all *C. acnes* strains, and approximately 18% of *C. acnes* strains carried bacteriophages [35].

Unlike phages from other bacterial species, the genomes of *C. acnes* bacteriophages show limited diversity despite their isolation from various geographical regions [25]. Marinelli et al. and Liu et al. sequenced bacteriophage genomes isolated from *C. acnes* and found a high degree of nucleotide identity between them [25,36]. Liu et al. also discovered identical bacteriophage strains among closely related individuals, suggesting the possibility of virus transmission from human to human [25]. In our study, we sequenced the genome of the bacteriophage PAP 1-1 isolated from *C. acnes*. OrthoANI analysis revealed that PAP 1-1 shares high nucleotide identity (87–93%) with other bacteriophage genomes. Phylogenetic analysis demonstrated close relatedness among bacteriophage genomes, and PAP 1-1 formed a clade with genomes isolated from *C. acnes* and/or *P. acnes*, clearly distinct from genomes of bacteriophages isolated from *Propionibacterium freudenreichii* strains (Figure 2). The genomic relatedness observed in the phylogenetic tree also suggests that PAP 1-1 shares common genes with bacteriophages ATCC29399BC and PAS50. *C. acnes* bacteriophage genomes harbor genes for endolysins, which contribute to bacterial cell wall degradation through the action of muramidases, amidases, endopeptidases, glucosaminidases, and transglycosylases [29,37,38]. Marinelli et al. proposed that these endolysins play a crucial role in using bacteriophages to treat acne infections [29]. The genes encoding these phage endolysins are highly conserved, with approximately 95% amino acid identity [36]. Phage endolysins have demonstrated efficacy in vitro and in vivo against various microbes [39]. In our study, we identified a gene encoding an endolysin through annotation using Prokka (Table 1). Future research could explore the use of *C. acnes* bacteriophage endolysins for treating *C. acnes* infections on the skin, potentially through genetic engineering, as previous studies have shown that bacteria remain sensitive to endolysins even after developing phage resistance [40].

Bacteriophage therapy has gained significant attention among scientists due to its successful treatment of infections in humans [41,42]. Bacteriophages have the ability to target both Gram-positive and Gram-negative bacteria, including multidrug-resistant strains [43,44,45,46]. Unlike antibiotics, which face increasing resistance over time, bacteriophages retain their ability to kill bacteria [47]. Moreover, since bacteriophages are host-specific, they exert minimal pressure on non-targeted microbial flora, unlike antibiotics. Several studies have investigated the specificity of bacteriophages against *C. acnes* [48,49]. For instance, Brown et al. isolated bacteriophages and demonstrated their ability to lyse *P. acnes* strains specifically, without affecting other members of the *Cutibacterium* genus [48]. In our present study, we also isolated the bacteriophage PAP 1-1 from acne lesions of volunteers and tested its efficacy against *C. acnes* KCTC 3314 in a dose-dependent manner (Figure 3). We observed that all of the tested concentrations of PAP 1-1 exhibited activity against *C. acnes* KCTC 3314. Among these concentrations, we selected 0.01 MOI to evaluate its combination with a crude bacteriocin from *L. lactis* CJNU 3001 and the commercially available nisin (Figure 4 and Figure 5). Bacteriocins have been previously reported to have antimicrobial effects and are recognized as effective bio-preservatives, such as nisin [23]. In our study, the bacteriocin treatment against *C. acnes* KCTC 3314 significantly reduced bacterial growth, and cell counts gradually decreased with increasing bacteriocin concentration (from 6.25 to 50 AU/mL) (Figure 4A). The co-treatment of the bacteriocin and 0.01 MOI resulted in a reduction in *C. acnes* KCTC 3314 cell counts to a certain level after 24 h of incubation, with a noticeable reduction observed in the co-treatment of 0.01 MOI and 50 AU/mL of the bacteriocin (Figure 4B). Similarly, regarding nisin activity, concentrations ranging between 12.5 IU/mL and 25 IU/mL reduced cell counts to a certain level compared with concentrations of 6.25 IU/mL and 3.125 IU/mL of nisin (Figure 5A). Notably, in the co-treatment of 0.01 MOI and 6.25 IU/mL of nisin, a significant reduction in the cell counts of *C. acnes* KCTC 3314 was observed (Figure 5B).

## 4. Materials and Methods

### 4.1. Isolation of Bacteriophage

To isolate the bacteriophage from the acne skin of volunteers, it was safely swapped twice with a sterile cotton swab, put into RCM (reinforced clostridial medium) broth inoculated to about 1 × 10^5^ CFU/mL of *C. acnes* KCTC (Korean Collection for Type Cultures) 3314 strain and cultured anaerobically at 37 °C for 72 h. The potential lysate was centrifuged at 10,000 rpm for 10 min at 4 °C, and the supernatant was recovered and filtered (0.45 μm syringe filter; Anylab Co., Seoul, Korea). To detect phage activity based on plaque formation, the method described by Liu et al. [33] was used. Briefly, 50 μL of the diluted supernatant was added to 3 mL of RCM soft agar (1.2% agar, *w/v*) where *C. acnes* KCTC 3314 strain had been inoculated to about 1 × 10^5^ CFU/mL, overlaid on an RCM agar plate and was incubated anaerobically at 37 °C for 48 h. After incubation, the formation of phage plaques was observed. Single phage plaque was picked up and inoculated to RCM broth where *C. acnes* KCTC 3314 strain had been inoculated and incubated anaerobically at 37 °C for 72 h. The culture was centrifuged at 10,000 rpm for 10 min at 4 °C to recover the supernatant, and it was filtered (0.45 μm syringe filter; Anylab Co.). To enumerate the number of plaques of the filtrate, it was decimally diluted and loaded on the RCM agar plate where *C. acnes* KCTC 3314 strain had been overlaid and anaerobically incubated at 37 °C for 48 h. The plaques were enumerated, and the number of plaques was represented as PFU (plaque-forming units) /mL. 

### 4.2. Transmission Electron Microscopy of PAP 1-1 Bacteriophage

A PEG solution (20% polyethylene glycol (PEG 8000), 2 M NaCl) was treated with the filtrate harboring the PAP 1-1 bacteriophage with a ratio of 1:1 and reacted on ice for 12 h. After that, the supernatant was completely removed via centrifugation with 12,000 rpm for 30 min at 4 °C, and the precipitate was suspended in sterilized SM buffer (100 mM NaCl, 8 mM MgSO_4_·7H_2_O, 50 mM Tris-HCl (pH 7.5), 0.01% gelatin), centrifuged at 13,000 rpm for 1 min at 4 °C, filtered (0.45 μm syringe filter, Anylab Co.), and used for transmission electron microscopy (TEM). Five microliters of the concentrated bacteriophage solution were dripped onto the grid and washed 3 times with water. After staining with 2% uranyl acetate, it was analyzed using a field emission transmission electron microscope (FE-TEM; JEM 2100F, Jeol, Japan) at the Korea Basic Science Institute (Chuncheon Center, Chuncheon, Korea).

### 4.3. Optimal Medium for Bacteriophage Propagation

To identify the growth medium that gives the most optimal condition for phage production, two different media were tested: RCM and BHI. Specifically, the bacteriophage PAP 1-1 was incubated (anaerobic condition at 37 °C) with *C. acnes* KCTC 3314 strain at MOIs of 0.01 or 0.1 separately in the two testing media. Sampling was carried out at 0, 3, 6, 9, 12, 18, 24, 48, and 72 h, and the double-layer agar method was used for phage enumeration.

### 4.4. Genomic DNA Extraction, Sequencing, and Annotation

To analyze the whole-genome sequence, genomic DNA was extracted by using a phage DNA isolation kit (Norgen Biotek Co., Thorold, ON, Canada), according to the manufacturer’s guide. The library was developed according to the TruSeq nano DNA library prep guide (Illumina, Inc., San Diego, CA, USA). Indexed libraries were then submitted to Illumina Hiseq X-10 (Illumina), and paired-end (2 × 151 bp) sequencing was performed by the Macrogen Co. (Seoul, Korea) for whole-genome sequencing. After sequencing, FastQC (v0.11.6) [50] was performed in order to check the read quality. To mitigate biases in the analysis, an internally developed script was employed to exclude the reads where the proportion of bases below Q20 was greater than 10%. Trimmomatic (v0.36) [51] was used to remove adapter sequences. First, de novo assembly was performed using various k-mer values in Spades (v3.12) [52]. The best contig was selected based on the status of the assembled results such as the number of contigs, contigs sum, and N50. Once the assembly was completed, we employed our in-house script to verify whether the ends of the contigs overlapped and subsequently connected them to form a circular contig. For gene prediction and annotation, Prokka (v1.12b) [53] was used, and for additional annotation, the assembled sequences were searched against GenBank non-redundant (NR) database, UniProt, GO, InterPro, Pfam, CD, TIGRFAM, and EggNOG using BLAST (v2.6.0+) [54].

### 4.5. Phylogenetic Analysis and Genomic Comparison

For the phylogenomic analysis, we retrieved the genomes of PAP 1-1 and closely related *Propionibacterium* bacteriophage genomes from the EMBL public database (https://www.ebi.ac.uk/genomes/phage.html, accessed on 03 April 2023). These genome sequences were aligned using ClustalW, and a phylogenetic tree based on the entire genome sequences was constructed with bootstrap values (1000 replications). The maximum likelihood (ML) tree was constructed in the MEGA7 software, utilizing the Kimura two-parameter model, the nearest-neighbor interchange heuristic search method, and partial deletion options [55].

Based on the phylogenomic tree, we selected the genomes of *P. acnes* ATCC29399BC and *P. acnes* PAS50 for genomic comparison with the PAP 1-1 genome. To perform this comparison, we employed the Proksee server (https://beta.proksee.ca/, accessed on 03 April 2023) to create circular alignments of sequence reads to the reference genomes.

### 4.6. Phage Lytic Activity of Bacteriophage PAP 1-1

In order to measure the phage lytic activity, it was incubated with *C. acnes* KCTC 3314 strain at four different MOIs, anaerobically at 37 °C [25]. Sampling was carried out at 0, 12, and 24 h to enumerate the viable cells (CFU/mL). Specifically, serial dilutions of the incubation mixture were performed 10 times, and then 10 µL of each dilution was spread on the RCM agar plate in triplicate. After incubation at 37 °C for 24 h, colonies were counted, and the values (CFU/mL) were calculated using the formula (total colonies × dilution factor × 100).

### 4.7. Bacteriocin Extraction and Purification

To assess the antibacterial activity of a bacteriocin derived from *L. lactis* CJNU 3001, a modified acetone extraction method was employed for partial purification [56]. Specifically, the supernatant obtained from an overnight culture was subjected to centrifugation at 8000 rpm for 10 min at 4 °C. The resulting supernatant was then filtered using a 0.45 μm membrane filter (Merck Millipore Co., Tullagreen, Ireland). Subsequently, the filtrate was concentrated by a factor of 10 using a rotary vacuum concentrator. To facilitate extraction, three portions of acetone were added to the concentrate, which was then placed at −20 °C for 3 h with intermittent shaking every 30 min. After completion of the extraction, centrifugation was carried out at 9000 rpm for 20 min at 4 °C to remove the supernatant and lower layer, leaving behind the middle layer. This middle layer was subsequently diluted with water to prepare a solution containing the bacteriocin, which was then stored at −20 °C until further use.

### 4.8. Combination Treatment with Bacteriophage PAP 1-1 and Crude Bacteriocin from Lactococcus Lactis and Nisin

The bacteriocin activity was measured as follows: The RCM soft agar inoculated with 100 μL culture of 1 × 10^5^ CFU/mL of *C. acnes* KCTC 3314 strain was poured on the RCM agar plate. The two-fold serial dilution of bacteriocin suspension was carried out, and then 2 μL suspension was spotted on the plate and anaerobically cultured at 37 °C for 48 h. The antibacterial activity of the bacteriocin was quantified using arbitrary units (AUs), where the AU/mL value was determined as the reciprocal of the maximum dilution factor at which the antibacterial activity is observed, multiplied by a conversion factor for 1 mL [57]. Subsequently, the antibacterial activity of various concentrations of the bacteriocin (0, 6.25, 12.5, 25, and 50 AU/mL) was assessed. To measure the antibacterial activity, each concentration of the bacteriocin was added to RCM broth containing *C. acnes* KCTC 3314 strain, which had been previously inoculated to a concentration of approximately 1 × 10^5^ CFU/mL, and the number of viable cells was then measured at different time points (0, 12, and 24) h.

To the mixture of bacteriophage PAP 1-1 and the *C. acnes* KCTC 3314 strain at MOI of 0.01, bacteriocin was added at different final concentrations of 0, 6.25, 12.5, 25, and 50 AU/mL, and the mixture was incubated anaerobically at 37 °C. Sampling was performed at 0, 12, and 24 h to enumerate the viable cells (CFU/mL). The antibacterial activity of nisin (Galacin Nisin 101, Galactic, Hainaut, Belgium) separately, at the different concentrations (0, 3.125, 6.25, 12.5, and 25 IU/mL) and with a combination of bacteriophage PAP 1-1, was determined against the *C. acnes* strain of KCTC 3314 in RCM broth. The viable cells were enumerated (CFU/mL) using the method described previously.

### 4.9. Statistical Analysis

All experiments were repeated three times, and the results are expressed as mean ± standard deviation. For statistical analysis, SPSS ver. 25 (Statistical Package for Social Sciences, SPSS Inc., Chicago, IL, USA) was used, and significance was verified using one-way ANOVA, and it was post-tested using Duncan’s multiple range test (*p* < 0.05).

## 5. Conclusions

The bacteriophage PAP 1-1, isolated and purified from acne lesions on the facial skin of volunteers, exhibited *siphovirus* morphology based on TEM observation. Phylogenetic analysis revealed that PAP 1-1 formed a distinct lineage alongside other bacteriophages isolated from *P. acnes* species. Genomic analysis indicated the presence of genes associated with lysis, including the endolysin gene responsible for host bacterium lysis. PAP 1-1 efficiently inhibited the growth of *C. acnes* KCTC 3314, an acne-causing bacterium isolated from human facial skin lesions. When combined with the crude bacteriocin from *L. lactis* CJNU 3001 and nisin, it showed a combination effect on the control of *C. acnes* KCTC 3314. Thus, the findings suggest that PAP 1-1 could serve as a promising candidate for acne infection control in combination with other drugs and/or antibiotics.

## Figures and Tables

**Figure 1 antibiotics-12-01035-f001:**
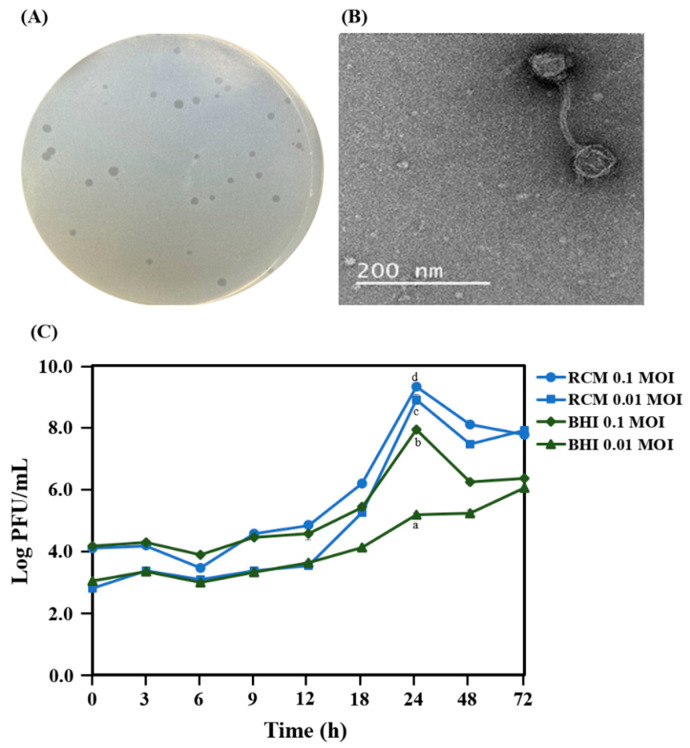
Formation of plaques (round, clear) in RCM agar plate (**A**) and phage particle morphology of the bacteriophage PAP 1-1 (**B**). Propagation of bacteriophage PAP 1-1 in broth (**C**). Data are provided as the mean value ± SD, measured in triplicate. Different letters present significant differences (*p* < 0.05).

**Figure 2 antibiotics-12-01035-f002:**
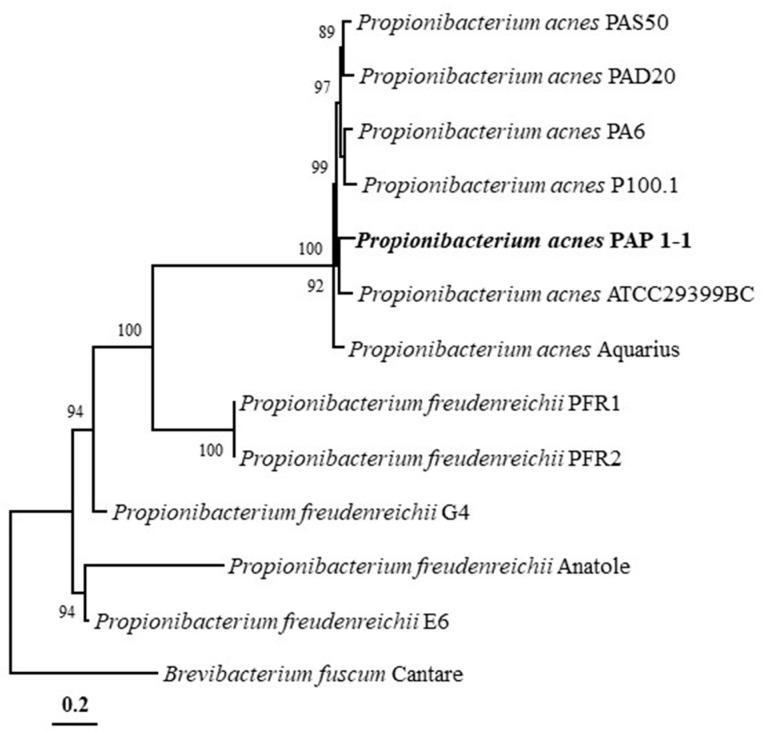
A maximum likelihood tree based on whole-genome sequences showing the phylogenetic relationships between PAP 1-1 and related bacteriophages. Bootstrap values are shown on nodes in percentages of 1000 replicates, with only values over 70%. *Brevibacterium fuscum* Cantare was used as an outgroup. The scale bar equals 0.2 changes per nucleotide position.

**Figure 3 antibiotics-12-01035-f003:**
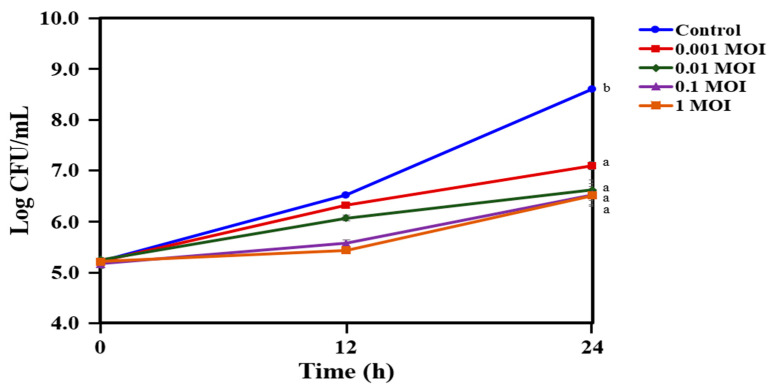
Reduction in the growth of *Cutibacterium acnes* KCTC 3314 cultured with bacteriophage PAP 1-1 ranging from 0.001 to 1 MOI. Data are provided as the mean value ± SD, measured in triplicate. Different letters present significant differences (*p* < 0.05).

**Figure 4 antibiotics-12-01035-f004:**
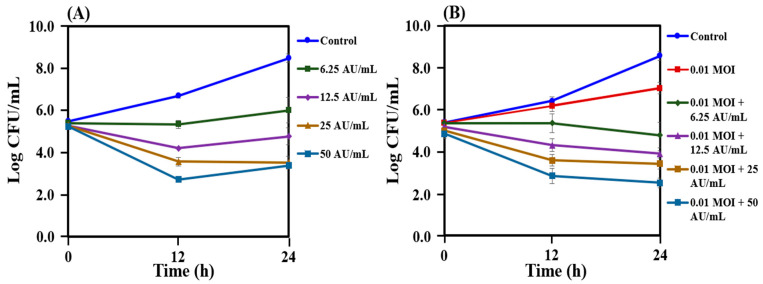
Reduction in the growth of *Cutibacterium acnes* KCTC 3314 cultured with crude bacteriocin from *L. lactis* CJNU 3001 alone (**A**) and a combination of the bacteriocin and bacteriophage PAP 1-1 (**B**). Data are provided as the mean value ± SD, measured in triplicate.

**Figure 5 antibiotics-12-01035-f005:**
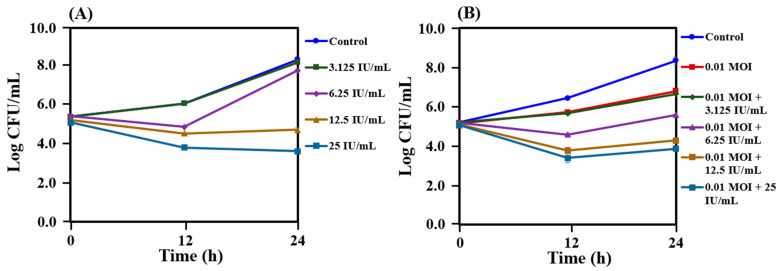
Reduction in the growth of *Cutibacterium acnes* KCTC 3314 cultured with nisin alone (**A**) and a combination of nisin and bacteriophage PAP 1-1 (**B**). Data are provided as the mean value ± SD, measured in triplicate.

**Table 1 antibiotics-12-01035-t001:** Lists of genes of bacteriophage PAP1-1 identified from genomic sequences.

CDs			Annotation
PAP1-1-1	47	373	Terminase small subunit
PAP1-1-2	378	1889	Gp2 (*Propionibacterium* phage PA6)
PAP1-1-3	1886	3211	Capsid and scaffold protein
PAP1-1-4	3215	3970	Capsid maturation protease
PAP1-1-5	4078	4638	Putative head assembly scaffold
PAP1-1-6	4645	5592	Major capsid protein
PAP1-1-7	5636	6097	Head-to-tail adaptor
PAP1-1-8	6099	6446	Gp8
PAP1-1-9	6452	6742	Gp9 (*Propionibacterium* phage PA6)
PAP1-1-10	6739	7110	Gp10 (*Propionibacterium* phage PAS50)
PAP1-1-11	7162	7794	Major tail protein (*Propionibacterium* phage P9.1)
PAP1-1-12	7822	8118	Gp12
PAP1-1-13	8217	8504	Putative tail assembly chaperone
PAP1-1-14	8513	11,278	Tape measure protein
PAP1-1-15	11,294	12,235	Minor tail protein
PAP1-1-16	12,243	13,400	Putative protease
PAP1-1-17	13,421	14,239	Putative minor tail protein
PAP1-1-18	14,540	15,328	Putative minor tail protein
PAP1-1-19	15,378	16,238	Putative endolysin
PAP1-1-20	16,251	16,649	Gp21
PAP1-1-21	16,735	17,250	Uncharacterized protein
PAP1-1-22	17,256	17,654	Helix–turn–helix DNA-binding domain protein
PAP1-1-23	17,658	17,942	Gp25 (*Propionibacterium* phage PAS50)
PAP1-1-24	17,954	18,274	Helix–turn–helix DNA-binding domain protein
PAP1-1-25	18,284	19,330	Gp27
PAP1-1-26	19,327	19,521	Gp28 (*Propionibacterium* phage PA6)
PAP1-1-27	19,505	20,062	Gp29.1 (*Propionibacterium* phage PAS50)
PAP1-1-28	20,059	20,625	Gp30 (*Propionibacterium* phage PA6)
PAP1-1-29	20,670	21,341	DNA primase
PAP1-1-30	21,539	21,895	Holliday junction resolvase
PAP1-1-31	21,895	22,758	DNA primase/helicase
PAP1-1-32	22,804	23,268	Gp35 (*Propionibacterium* phage PAS50)
PAP1-1-33	23,321	23,731	Gp36 (*Propionibacterium* phage PAS50)
PAP1-1-34	23,728	24,675	Cas4 family exonuclease (*Propionibacterium* phage Pirate)
PAP1-1-35	24,675	25,028	Gp38 (*Propionibacterium* phage PAS50)
PAP1-1-36	25,134	25,337	Gp39
PAP1-1-37	25,334	25,561	Gp40 (*Propionibacterium* phage PAS50)
PAP1-1-38	25,578	26,111	Gp41 (*Propionibacterium* phage PAS50)
PAP1-1-39	26,232	26,543	Gp43 (*Propionibacterium* phage PA6)
PAP1-1-40	26,578	26,886	Gp43 (*Propionibacterium* phage PA6)

## Data Availability

Data are available in a publicly accessible repository and within the article.

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
