# Peer review of "Cutibacterium acnes* KCTC 3314 Growth Reduction with the Combined Use of Bacteriophage PAP 1-1 and Nisin"

_antibiotics, 2023, doi:10.3390/antibiotics12061035_

Round 1
Reviewer 1 Report
In this manuscript titled 'Combination treatment for inhibition of the growth of Cutibacterium acnes with bacteriophage PAP 1-1 and bacteriocin', the authors have presented their findings about isolation and characterization of a phage PAP 1-1 as well as it’s in vitro growth inhibition activity against C. acnes by itself as well as in combination with a bacteriocin.
The authors do a good job in presenting why phages could be used an alternative solution for treating acne and show how the phage they isolated has the potential activity in vitro to achieve the same. The authors successfully isolated genomic DNA from the phage, sequenced and were able to annotate it with information from other closely related phages. The authors also present data about growth inhibition activity of the phage in combination with a crude bacteriocin and commercially available Nisin.
However, there is a lot of room for improvement in the manuscript, especially in the presentation of data, use of language to present the data and the design of a few experiments as well. Extensive comments have been provided in the attached pdf, pointing out to grammatical issues and other suggestions to add/remove statements that could make the submission better. To highlight a few,
1. The introduction needs a better background for the bacteriocins.
2. The genome analysis of the phage needs to be elobarated more. Talk about the functional annotations assigned to the phage genes.
3. Figures could be organized in a better way and also need to be zoomed in/ higher quality in a few cases. Make sure the legend or text in the figures is legible and readable. The use of alphabets to indicate significant differences is a little confusing, the authors could probably avoid them and still the readers would understand the significant differences wherever they are present.
4. The experiments with crude bacteriocin and Nisin could use a few more data points with different phage MOIs as suggested in the pdf. It will help understand the individual contributions of the phage and the bacteriocin to the inhibition activity.
5. This is not necessarily a strong concern, but could the authors use a negative control for the bacteriocin experiments? As in, explain why choose the bacteriocin from Lactococcus lactis? Would bacteriocins from other sources also work or would they not? If the specificity to Lactococcus bacteriocin is real, then that information needs to be incorporated into the title of the paper.
6. Regarding supplimentary information, figure S1 does not provide any useful info that adds to the paper, other than the statement that has already been made in the paper. There seems to be a region with very low GC content, that could potentially be the origin of replication for the phage, if information like that can be added, it might justify keeping this image. \
7. Figure S2 needs to be larger, the GC skews are barely visible and do not really give any additional value to the reader. Also there are 6 panels/sectors for the ORFs (blue), which are probably ORFs in the 6 different translation frames? But which of these are the final ORFS annotated by the authors? what is their location on the genome? Would also suggest making it a linear figure, rather than a circular presentation may be.
8. Table S1 could actually be moved into the main article as it provides a lot of useful information. It is also clear that the phage has different modules responsible for head assembly, tail assembly, lysis and DNA replication/transcription. This information should be projected somewhere in the results or discussion section of the article.

There are a lot of grammatical errors spread across the manuscript. It is highly recommended that the authors run their revised manuscript through grammar checking tools such as 'Grammarly' before submitting it.
Author Response
Responses to reviewer #1
1. Line 70-106, The introduction needs a better background; We have described bacteriocins background as suggested.
2. Line 135-136; 223-224; The genome analysis of the phage; We have added information about essential functional genes.
3. Figures could be organized in a better; We have modified the figures as suggested by the reviewer.
4. The experiments with crude bacteriocin and Nisin could use a few more data points with different phage MOIs as suggested in the pdf. It will help understand the individual contributions of the phage and the bacteriocin to the inhibition activity; We agree with the comment but here we just wanted to show the combination effect to efficient control the pathogen.
5. This is not necessarily a strong concern, but could the authors use a negative control for the bacteriocin experiments? As in, explain why choose the bacteriocin from Lactococcus lactis? Would bacteriocins from other sources also work or would they not? If the specificity to Lactococcus bacteriocin is real, then that information needs to be incorporated into the title of the paper.; We considered only the bacteriocin from a Lactococcus lactis strain since it produces nisin which is commercially available and used as a biopreservative to foods worldwide. To specify, we added the name on the title.
6. Regarding supplimentary information, figure S1; Figure S1 deleted as suggested.
7. Figure S2 needs to be larger; We have deleted the figure to remove any further confusion.
8. Lines135-136, 224; Table S1 could actually be moved into the main article; Table S1 moved the main article, additionally some of the important information about genes and genome has also been described as suggested by the reviewer.
Reviewer 2 Report
The manuscript entitled “Combination treatment for inhibition of the growth of Cutibacterium acnes with bacteriophage PAP 1-1 and bacteriocin” described the isolation and characterization of C. acnes phage. Morphology, genome, and phylogenetic relationships of phage PAP 1-1 were also investigated. Phage was also tested to kill C. acnes by mixing with bacteriocins from L. lactis and commercial nisin.
This MS is well-written and has scientific soundness, and generates a significant impact on the fields. However, there are some flaws that must be revised. Here is my suggestion:
Line 71: The action mode of bacteriocin on pathogenic bacteria should be included in the Introduction section.
Section 2.1: plaque morphotype (halo, turbid or clear plaque???) and size of plaque in diameter should be added. The size of the phage’s head and tail via TEM study should be measured (width x length) and added.
Figure 1B: kindly replace this figure with a higher magnification.
Section 2.2: Optimum medium and phage genome should be separated.
Section 2.2: Figure S2 should be added to the main manuscript instead of a supplementary file. The position/location of essential genes (or CDS) presented in the phage genome (not all genes) should be indicated.
Section 2.4-2.6 and Section 4.6-4.8: The number of phage concentrations during the experiments was not determined. These facts hamper the interpretation of the results. Kindly add the result and interpret/discuss it along with the previous works reported by other authors.
Line 267-270: this sentence should be revised: In order to purify the isolated phage…….
Line 288-289: the concentration of uranyl acetate should be added.
Line 290: the magnification of the TEM study should be added.
Line 325: URL should be added.
Kindly check the format of the references.
Kindly check Adv., Adj. conjunction, articles with the appropriate words, and check the symbols for example "comma", "colon", "semicolon", and "full-stop".
Author Response
Responses to reviewer #2
1. Line 70–106; The action mode of bacteriocin on pathogenic bacteria; We have included the information about mode of bacteriocin in the manuscript as suggested by the reviewer.
2. Section 2.1: plaque morphotype (halo, turbid or clear plaque???) and size of plaque in diameter should be added. The size of the phage’s head and tail via TEM study should be measured (width x length) and added; We newly made the plaques figure for clearance and amplified the TEM image to estimate the size of the phage.
3. Figure 1B: kindly replace this figure with a higher magnification; Unfortunately we did not get the image with a higher magnificant. Instead of it, we amplified the TEM image to estimate the size of the phage.
4. Lines 123–139; Section 2.2: Optimum medium and phage genome should be separated; Separated as suggested.
5. Section 2.2: Figure S2 should be added to the main manuscript; We have deleted the figure to remove any further confusion.
6. Lines 160–169; Section 2.4-2.6 and Section 4.6-4.8: The number of phage concentrations; We have updated the informated as suggested by the reviewer.
7. Lines 293–297; In order to purify the isolated phage; Rephrased as suggested.
8. Lines 315–318; the concentration of uranyl acetate should be added; Concentration added as suggested.
9. Figure 1; the magnification of the TEM; Instead of a magnification, a scale bar is shown in the Figure.
10. Lines 349–351; Added as suggested.
Reviewer 3 Report
The provided manuscript is about “Combination treatment for inhibition of the growth of Cutibacterium acnes with bacteriophage PAP 1-1 and bacteriocin”.
The phage PAP 1-1 was isolated against C. acnes strain KCTC 3314; phage lytic activity, separately and with combination of bacteriocin, was studied just on one host (isolating) strain. Phage particle morphology was characterized by EM; and its genome was sequenced/analyzed and phylogenetic tree was developed. But search for lysogenic genes among the annotation were not performed; and just clear plaque morphology is considered as an “indicator” for “lytic phage” nature/type.
Searching for solutions for antimicrobial resistance is indeed real challenge worldwide.
But to judge the work as a novel or significant enough to be published is difficult as it is kind of reporting of one phage isolation, that wasn’t characterized completely to evaluate its therapeutic potential. At least it is not well explained and needs to be improved.
Introduction is very general and doesn’t cover phages against C. acnes.
Discussion is almost repetition of the results.
Conclusion are not developed well.
Methods are not described in a standardized way (especially similar ones): sometimes extra details are given (even repeated several times), sometimes important data is missing (phage pfu/ml used, bacterial cfu/ml used, phage concentration is confused with MOI).
I’d like the authors to address the main suggestions/comments:
Pages 2-3: “Combination treatment for inhibition of the growth of Cutibacterium acnes with bacteriophage PAP 1-1 and bacteriocin”
- Suggest to change the title something like that:” Cutibacterium acnes KCTC 3314 growth reduction with combination using of bacteriophage PAP 1-1 and bacteriocin” – as “inhibition” is not right term in this case and work was done just on one strain.
Pages 14-15: “bacteriophage therapy can be a suitable alternative method to antibiotics.”
- phages can’t replace antibiotics, so suggest to write: bacteriophage could be used in synergy or combination with antibiotics/antimicrobials instead of.
Pages 17-18: “Morphology un-17 der TEM illustrate to classify this newly isolated phage in Siphoviridae family”.
- Morphology classification is not used anymore. Please rephrase the sentence based on the current classification system:
https://doi.org/10.1007/s00705-022-05694-2
Page 20: “bacteriophages isolated from species Propionibacterium acnes (former name of C. acnes)”
- Do you mean phages were isolated from bacterial strain? Or phages active to Propionibacterium acnes?
Pages 42-44: “The using of these antibiotics or the antibiotics showing strong action against C. acnes strains have a main concern toward modification of other skin microbiota.”
- The sentence needs to be rephrased or changed. The first part of sentence is confusing; plus the main reason of changing skin microbiota is antibiotic-resistance as well, that is already postulated in the previous sentence of pages 40-41.
Pages 73-76: explain why was KCTC 3314 selected, importance of it. Provide scientific details of the strain (reference).
Pages 80-85: remove this from the section of results. It is described in methods.
Pages 86-48:
- Write “Siphovirus morphology” as classification based on morphology is not used anymore: “Bacteriophage PAP 1-1 have the morphology resemblance to the members of the family Siphoviridae”- rephrase the sentence.
Pages 90-91:
- write phage plaque morphology instead: “Plaques formation in RCM agar plate”. At the same time, according to the picture, plaques don’t look like homogenous. There is quite big difference in plaques’ diameters.
- phage particle morphology instead of “and morphology by TEM”
- “with icosahedral head and tail of the bacteriophage PAP 1-1” - this is already written in the text above.
Pages 95-97: “2.2. Optimum Medium and Genome Features”- split this sub-section in two.
- Remove this: “Bacteriophage PAP 1-1 was propagated in RCM and BHI broth where C. acnes KCTC 3314 was inoculated” as it is described in the section of methods; write just results.
Pages 97-106: again, it is repetition what is described in the section of methods; write just results.
Pages 113-125: again, write just results (considering image annotation as well).
Page 127-140:
- The term “inhibition is less used with phages, better to write phage activity study against … or bacterial reduction instead of “Growth inhibition of C. acnes KCTC 3314 with bacteriophage PAP 1-1”.
- “The bacteriophage PAP 1-1showed activity against C. acnes KCTC 3314 in comparison with control (Figure 4).”- this saying nothing. Just activity was already demonstrated. Move to the next sentence “as it was shown”.
- “0.001 MOI concentration of bacteriophage PAP 1-1” – MOI is not concentration. Write with MOI, phage revealed…
- “greatly control the growth of C. acnes KCTC 3314 to 7.0 log CFU/mL in comparison with control 8.6 log CFU/mL” –what does mean greatly control? It is just in 1.6 log reduction. Please rephrase it.
- “however with increasing concentration to 0.01 MOI of bacteriophage PAP 1-1”- again, rephrase it.
- resulted in two logs reduction instead of “a notable reduction in cell counts of C. acnes KCTC 3314 to log 6.6 CFU/mL.”
- “The gradual increase in concentration of bacteriophage PAP 1-1 from 0.1 to 1 MOI control the growth of C. acnes KCTC 3314 until 12 h of incubation that gradually increase to 6.5 CFU/mL similar to 0.01 MOI after 24 h of incubation.”- please, rephrase this as well in a laconic way.
- “Figure 4. MOI means multiplicity of infection as (PFU) / host (CFU)– remove this as it is given in the section of methods.
Page 165-186:
- Write growth reduction instead of “Growth inhibition of C. acnes KCTC 3314 followed by co-treatment with bacteriophage PAP 165 1-1 and nisin”
- “In our previous partial whole genome sequencing of L. lactis CJNU 3001 (GenBank accession no. JAIZVR000000000.1), it harbored corresponding structural gene for nisin and commercially available pure nisin was used for further analysis of co-treatment”- this should be jut in section of methods.
- Pages -172-179: please rephrase, give in log reductions and write in a way to be easy read.
- Remove this “MOI means multiplicity of infection as (PFU) / host (CFU)” as it is given in the section of methods.
Page 192: Suggest to write studied instead “checked activity”
Pages 193-195-: “The study has significantly increased our understanding of activity of bacteriophage PAP 1-1 against C. acnes KCTC 3314 isolated from acne lesion in human facial skin” – it is confusing, how was understanding increased, if phage was isolated and studied for the first time?
Pages 196-198: “TEM based morphological characterization showed that PAP 1-1 (Figure 1B) belongs to the Siphoviridae family as PAP 1-1 possess icosahedral head and long tail [25,26].” – again write Siphovirus morphology... and here it is a repetition of the results again.
Page 251: “All of the tested concentration of PAP 1-1 were found to have activity against C. acnes KCTC 3314.”- this is very generic and saying nothing. Lytic activity needs to be evaluated somehow. In your case bacterial reduction was max two logs.
Pages 242-255: again MOI is not concentration. And results are repeated, instead of do be discussed and compared some other data results.
Pages 259-277: « «Isolation of bacteriophage specific to Cutibacterium acnes
- Refer the method itself and what kind of modification was used.
- Remove “an” “an RCM broth inoculated with C. acnes KCTC 3314 strain”, - and refer the strain (KCTC (Korean Collection for Type Cultures).)
- “C. acnes KCTC 3314 strain,”- write what OD or cfu/ml was used.
- “RCM broth” - refer the media or indicate if it was lab made.
- “cultured anaerobically at 37°C for 72 h” – better to write was incubated and indicate what level of anaerobic condition and which tool was used and give its reference.
- centrifuged “at” instead “with 10,000 rpm”. Consider it further as well
- use “potential lysate” instead of “The culture”
- “The filtrate was spotted on RCM agar plate where C. acnes KCTC 3314 strain had been overlaid”- name phage detection method, add was it soft agar layer or direct one on solid agar, one spot of lysates was used or dilutions of spots?, and refer the method.
- “In order to purely isolated bacteriophages from the supernatant in which the plaque was confirmed” – sentence is confusing, please rephrase it. I guess you want to write how plaque purification was done?
- “soft agar (1.2% agar, w/v) where C. acnes KCTC 3314 strain had been inoculated” – add OD or cfu/ml that was used.
- “After recovering purely isolated plaque” – it is written that just one round of individual plaque passaging was used, that is not enough for phage plaque purification; and to state that pure phage culture (generation of one particle) was used further for phage lytic activity study.
- Name phage enumeration method.
Pages 293-299: “Optimal medium for bacteriophage propagation “
– Do you mean selection of medium for propagation? refer the method
– “added to RCM or BHI broth” - wat does mean “or”, was tested both?
– Method needs to be re-written otherwise it is difficult to understand.
Pages 303-304:
– write what concentration (pfu/ml and volume) of phage was used instead of “bacteriophage PAP 1-1 was grown in RCM 303 broth medium”.
Page 305-306:
– write library was developed instead of “Library preparations were performed”
Pages 310-311: sentences need to be re-phrased otherwise it is difficult to understand
Pages 314-315: sentences need to be re-phrased otherwise it is difficult to understand
Pages 319-320: “The genome sequence of PAP 1-1 319 was deposited in the database of GenBank (accession number OP491959)” – this should go to the section of results.
Pages 324-333: the section needs to be re-written in a more constructive way with less repetitions (tautology) and easy to read.
Page 335-342: Better to write bacterial reduction or phage lytic activity instead of “Antibacterial activity of bacteriophage PAP 1-1”. And please, refer the method.
Pages 337-339: the sentence is not correctly formulated.
– What does mean 0.001, 0.01, 0.1, 1 MOI of bacteriophage PAP 1-1? MOI is not related just to phage. The same was above as well. Needs to be written that three different MOIs or phage/bacteria ratio was used for study. And, if bacterial cfu/ml is mentioned then write phage pfu/ml as well that gives final MOIs or ratios…
– Remove this “After that”
– “The number of viable cells was measured by diluting it in 0.1% peptone water by the decimal dilution method and expressed as log CFU/mL” – describe how bacteria was cultured, with just dilution measuring cfu/ml not possible.
Page 344-371: “Combination treatment of bacteriophage PAP 1-1 and crude bacteriocin from Lactococcus lactis”- better to write treatment “with” instead “of”. or
– Better to split method in two sections: “bacteriocin extraction and purification” and “phage and bacteriocin activity study”.
– the section needs to be re-written in a more constructive way with less repetitions (tautology) and easy to read, especially the part of bacteriocin purification.
– What does mean “with 1% of culture of C. acnes KCTC 3314 strain” – better to write OD or cru/ml and volume of bacterial suspension used.
– Write two-fold serial dilution of bacteriocin suspension was done instead of “The bacteriocin solution was serially 2-fold diluted”
– “spotted on the plate” – write volume of spot used
– Phages 358-360: sentence needs to be rephrased to be easy read and understood.
– Phages 361-365: It is really confusing, was both: the agar and liquid method used for bacteriocin activity evaluation? Please rewrite it in a clear way and don’t repeat/describe the same methods used just with different factors.
Pages 373-382: include this section with one described above and again don’t repeat the same procedures.
- “the viable cell count was measured under anaerobic conditions over times (0, 12, 24 h) at 37°C.”- what does mean was measured under anaerobic condition? Do you mean was incubated under anaerobic condition and over times the samples were taken for cfu/ml counting?
Page 394-403:
- “Bacteriophage PAP 1-1 was successfully isolated and purified from acne lesions from facial skin of volunteers.” – this not conclusion. Plus, what does mean “successfully isolated”?
- “Based on morphology under TEM phage PAP 1-1 belongs to the Siphoviridae family” – correct it
- “The phylogenetic analysis showed that PAP 1-1 formed a phyletic lineage with the members of bacteriophages isolated from P. acnes species” – this is result not conclusion.
- “Genomic analysis showed PAP 1-1 carried endolysin gene that hypothesized the PAP 1-1 can be a suitable candidate against C. acnes” – this not enough for the conclusion.
- “PAP 1-1 was effective against C. acnes KCTC 3314 isolated from acne lesion in human facial skin”. How much effective? Needs to be evaluated to make kind of conclusion.
- In combination with crude bacteriocin from L. lactis CJNU3001 and nisin, a significant reduction was noticed of C. acnes KCTC 3314”. What is significant reduction? Needs to be evaluated to make kind of conclusion.
- “Therefore, PAP 1-1 could be used as a suitable candidate to control acne infection as an alternative to antibiotics” - First off all lytic activity study given here is not enough to evaluate phage therapeutic potential. Second, phages cannot replace antibiotics.
Moderate editing of English language
Author Response
Responses to reviewer #3
1. Lines 53–69; Introduction is very general; We have added the information as suggested by the reviewer.
2. Lines 215–279; Discussion is almost repetition of the results; Updated the discussion part as suggested by the reviewer.
3. Lines 425–436; Conclusion are not developed well; We have made amendments in conclusion parts.
4. Section 4; Methods are not described in a standardized way; We have made changes according to suggestions of the reviewers in material and method section.
5. Suggest to change the title; We have updated the title as suggested.
6. Lines 14-15; phages can’t replace antibiotics; We have updated as suggested by the reviewer.
7. Lines 18-19; Morphology classification is not used; Made ammendments as suggested.
8. Line 20-21; bacteriophages isolated from species; We have updated the information.
9. The sentence needs to be rephrased or changed; We have deleted the lines as reviewer has suggested that sentence is repetition of previous described concepts.
10. Line 111; Why was KCTC 3314 selected; We have added the information in the manuscript that this particular strain was selected to include in this study as KCTC 3314 is the type strain.
11. Lines 117–120; Remove this from results; Removed as suggested.
12. Line 120; Write Siphovirus morphology; Changed as suggested.
13. Lines 117–120; write phage plaque morphology; Updated as suggested.
14. Section 2.2; Split this sub-section; Splitted as suggested by the reviewer.
15. Line 120-121; Remove this; Removed as suggested.
16. Line 131-132; Repetition of the methods; Removed as suggested.
17. Lines 132–137; Write just results; Changed as suggested.
18. Line 173, 193…; The term inhibition; Replaced with reduction.
19. Line 162, 182; The bacteriophage PAP1-1 showed activity against; Made ammendments as suggested by all reviewers.
20. Write with MOI; Changed in the whole manuscript.
21. Line 293…; Write what OD or cfu/ml; Mentioned as suggested.
22. Line 294 better to write was incubated; Added as suggested.
23. Line 286; Potential lysate; updated as suggested.
24. Line 288; The filtrate was spotted; Updated as suggested.
25. Line 290–294; In order to purely; Rephrased as suggested.
26. Line 293; Soft agar; Added cfu/ml used in this study.
27. After recovering purely isolated plaque” – it is written that just one round of individual plaque passaging was used, that is not enough for phage plaque purification; and to state that pure phage culture (generation of one particle) was used further for phage lytic activity study; we deleted “purely” for clearance.
28. Line 291; Phage enumeration; updated as suggested.
29. Lines 320-323; Optimum medium; We have made ammendments in this section as suggested by the reviewer.
30. Lines 361–368; write what concentration; We have updated the information as suggested by the reviewers.
31. Lines 329-343; write library was developed instead of; Made amendments in the genome analysis section as suggested.
32. Lines 361–367; What does mean 0.001, 0.01, 0.1, 1 MOI; We have updated the information as suggested by the reviewers.
33. Lines 371–414; Better to split method in two sections; We have updated the information in this section as suggested by the reviewer.
34. Lines 426–435; Conclusion ammendments; We have rephrased and updated the information as suggested by the reviewer.
Round 2
Reviewer 2 Report
I have no further comments.
I have no further comments.
Author Response
Thanks for kind response.
Reviewer 3 Report
Pages 18-19: “Transmission electron microscopy (TEM) was employed to examine the morphology of the newly isolated phage and found Siphoviridae morphology”.
- Needs to be rephrase like this: Examination under Transmission electron microscopy (TEM) revealed that newly isolated phage has a morphology typical to Siphoviruses.
And further in the text use Siphoviruse as term “Siphoviridae” refers to Family and I already wrote previously, that morphology based classification is not used anymore.
Page 290: “The one spot of filtrate was spotted on RCM”
- Write one dilution spot
Pages 290-292:
“The one spot of filtrate was spotted on RCM agar plate where C. acnes KCTC 3314 strain had been overlaid and incubated anaerobically Antibiotics 2023, 12, x FOR PEER REVIEW 9 of 15 at 37°C for 48 h to check formation of plaques.”
- Remove this because it is written again further,
Pages 292-293:
“The bacteriophages was isolated from the supernatant in which the plaque was confirmed”
- It is very confusing. What do you mean isolated? Supernatant of what? Supernatant is liquid, how it is possible to see plaque in a liquid?
Instead write: to detect phage activity by plaque formation, the following the method described by Liu 293 et al [25] was used,
Page 297: “After recovering isolated plaque, it was inoculated in RCM broth where”
- Instead write: After incubation, phage plaques formation was observed. Single phage plaque was picked up and inoculated to…
Pages 320-325: “To enumerate better growth medium bacteriophage PAP 1-1 was added to RCM and/or BHI broth to be 0.01 or 0.1 MOI and then C. acnes KCTC 3314 strain was inoculated to about 1×105 CFU/mL. Then, the bacteriophage plaque number was measured over time (0, 3, 6, 9, 12, 18, 24, 48, 72 h) in anaerobic conditions at 37°C.”
Needs to be rephrased:
To identify the growth medium that gives the most optimal condition for phage production, two different media: RCM322 and BHI were tested. Particularly, bacteriophage PAP 1-1 was incubated (anaerobic condition at 37°C) with C. acnes KCTC 3314 strain at MOIs of 0.01 or 0.1 separately in two testing media. The sampling was done at 0, 3, 6, 9, 12, 18, 24, 48, 72 h and double layer agar method was used for phage enumeration.
Page 329: “0.1 MOI was grown in RCM broth medium” –remove this’ first of all it is not correct, that I wrote you previously; plus, here it doesn’t say anything.
Page 363-369: “In order to measure the phage lytic activity, four different MOIs were added to RCM broth where C. acnes KCTC 3314 strain had been inoculated to about 1×105 CFU/mL [25]. The number of viable cells was measured over time (0, 12, 24 h) during anaerobic culture at 37°C. The number of viable cells were measured as log CFU/mL. To measure CFU/mL, 10 times serial dilutions of culture was made, then we poured 10 µL of each dilution on RCM agar plate in triplicate form. After incubation at 37 ℃ for 24 h, colonies were counted 368 and determined CFU/mL using formula (total colonies × dilution factor × 100).”
- instead write: In order to measure the phage lytic activity, it was incubated with C. acnes KCTC 3314 strain at four different MOIs, anaerobically at 37°C [25]. The sampling was done at 0, 12, 24 h to enumerate viable cells (CFU/mL). Particularly, 10 times serial dilutions of incubation mixture were done, then 10 µL of each dilution was speeded on RCM agar plate in triplicate. After incubation at 37 ℃ for 24 h, colonies were counted and CFU/mL was calculated using formula (total colonies × dilution factor × 100).
Page 403-405: “bacteriophage PAP 1-1 was inoculated to 0.01 MOI in RCM broth where the bacteriocin had been added to 0, 6.25, 12.5, 25, 50 AU/mL, and then C. acnes KCTC 3314 strain was inoculated to about 1×105 CFU/mL. Thereafter, the combination effect was confirmed by measuring the number of viable cells over time (0, 12, 24 h) after incubation under anaerobic conditions at 37°C.”
- Instead write: To the mixture of bacteriophage PAP 1-1 and C. acnes KCTC 3314 strain at MOI of 0.01, was added bacteriocin at the different final concentration: 0, 6.25, 12.5, 25, 50 AU/mL and incubated anaerobically at 37 ℃. The sampling was done at 0, 12, 24 h to enumerate viable cells (CFU/mL).
Page 408-414: “For the measurement of antibacterial activity of nisin, the nisin (Galacin Nisin 101, Galactic, Belgium) was adjusted to different concentrations of 0, 3.125, 6.25, 12.5, and 25 IU/mL in RCM broth. These concentrations were then applied against the C. acnes KCTC 3314 strain, and the viable cell counts were measured using the method described previously. To assess the combined effect of bacteriophage PAP 1-1 and nisin, a MOI of 0.01 was inoculated in RCM broth. Prior to inoculation, nisin at concentrations of 0, 3.125, 6.25, 12.5 and 25 IU/mL was added to the broth. Subsequently, C. acnes KCTC 3314 was inoculated, and the viable cell counts were measured accordingly.”
- Instead write: the antibacterial activity of nisin (Galacin Nisin 101, Galactic, Belgium) separately, at the different concentrations (0, 3.125, 6.25, 12.5, 25 IU/mL) and with combination of bacteriophage PAP 1-1, was determined against the C. acnes strain of KCTC 3314 in RCM broth. The viable cells were enumerated (CFU/mL) using the method described previously.
Moderate editing of English language required
Author Response
We appreciate for the very kind and excellent comments from the reviewers. We have carefully revised the manuscript based on the comments and responded to all comments point-by-point as follows.
Responses to reviewer
1. Lines 18-19, Needs to be rephrase; We have rephrased as suggested by the reviewer.
2. Line 120, Further in the text use Siphoviruse; We have updated as suggested by the reviewer.
3. Lines 290-291, Remove this because it is written again further; We have removed the sentence as suggested.
4. Lines 290-291, Instead write; We have rephrased as suggested by the reviewer.
5. Lines 294-295; Instead write; We have rephrased as suggested by the reviewer.
6. Lines 320-324, Needs to be rephrased; We have rephrased as suggested by the reviewer.
7. Line 328, Remove this’ first of all it is not correct; We have removed as suggested.
8. Lines 362-367, Instead write; We have rephrased as suggested by the reviewer.
9. Lines 399-402, Instead write; We have rephrased as suggested by the reviewer.
10. Lines 402-406, Instead write; We have rephrased as suggested by the reviewer.
Round 3
Reviewer 3 Report
the previous comments/suggestions are followed.
Moderate editing of English language required
spelling needs to be checked